# Mining the Benefits of Two-stage and One-stage HOI Detection

**Aixi Zhang**[1]* **Yue Liao**[2]* **Si Liu**[2]†
**Miao Lu**[1] **Yongliang Wang**[1] **Chen Gao**[2] **Xiaobo Li**[1]
[1]Alibaba Group    [2]Beihang University

## Abstract

Two-stage methods have dominated Human-Object Interaction (HOI) detection for several years. Recently, one-stage HOI detection methods have become popular. In this paper, we aim to explore the essential pros and cons of two-stage and one-stage methods. With this as the goal, we find that conventional two-stage methods mainly suffer from positioning positive interactive human-object pairs, while one-stage methods are challenging to make an appropriate trade-off on multi-task learning, *i.e.*, object detection, and interaction classification. Therefore, a core problem is how to take the essence and discard the dregs from the conventional two types of methods. To this end, we propose a novel one-stage framework with disentangling human-object detection and interaction classification in a cascade manner. In detail, we first design a human-object pair generator based on a state-of-the-art one-stage HOI detector by removing the interaction classification module or head and then design a relatively isolated interaction classifier to classify each human-object pair. Two cascade decoders in our proposed framework can focus on one specific task, detection or interaction classification. In terms of the specific implementation, we adopt a transformer-based HOI detector as our base model. The newly introduced disentangling paradigm outperforms existing methods by a large margin, with a significant relative mAP gain of $9.32\%$ on HICO-Det. The source codes are available at https://github.com/YueLiao/CDN.

## 1 Introduction

The goal of Human-Object Interaction (HOI) detection [2, 20, 6, 18, 7, 8, 17, 3] is to make a machine detailedly understand human activities from a static image. Human activities in this task are abstracted as a set of <human, object, action> HOI triplets. Thus, an HOI detector is required to locate human-object pairs and classify their corresponding action simultaneously. Based on this definition, we can summarize conventional HOI detection methods into two paradigms, *i.e.*, two-stage methods, and one-stage methods. These two paradigms have made significant progress with the development of deep learning, but both paradigms still have their shortcomings due to their structural design. This paper aims to present a detailed analysis of methods under these two paradigms and propose a solution to mine the benefits of two-stage and one-stage methods.

We first take a closer look at the conventional two-stage and one-stage HOI detectors. Conventional two-stage methods [6, 2, 18, 5] are mostly with a serial architecture. As shown in Figure 1 (a), two-stage methods detect humans and objects first and then feeds the human-object pairs, which are generated by matching humans and objects one by one, into an interaction classifier. The serial architecture suffers from locating the interactive human-object pairs under the interference of a large number of negative pairs only based on local region features. Otherwise, the efficiency of

---

*Equal contribution
†Corresponding author (liusi@buaa.edu.cn)

35th Conference on Neural Information Processing Systems (NeurIPS 2021).

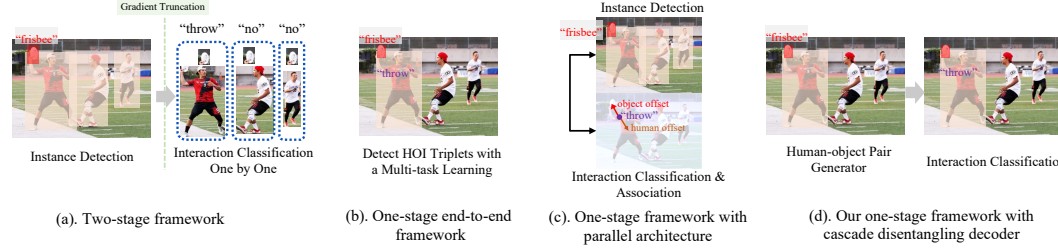

Figure 1: (a) Two-stage framework, (b) one-stage end-to-end framework, (c) one-stage framework with parallel architecture, and (d) our one-stage framework with a cascade disentangling head.

two-stage methods is also limited by the serial architecture. To alleviate these problems, one-stage methods [20, 13, 39, 3, 28, 14] are proposed to detect the HOI triplets directly and break HOI detection as multi-task learning, *i.e.*, human-object detection and interaction classification, which is shown in Figure 1(b). Therefore, one-stage methods can easily focus on the interactive human-object pairs and effectively extract corresponding features in an end-to-end manner. However, it is difficult for a single model to make a good trade-off on multi-task learning since human-object detection and interaction classification are two very different tasks, which requires the model to focus on different visual features. As shown in Figure 1(c), though some previous methods [20, 3] design two parallel branches to detect instances and predict interaction respectively, the interaction classification branch still needs to regress additional offsets to associate humans and objects. Thus the interaction branch is also required to make a trade-off between interaction classification and human and object positioning.

Therefore, the intuitive idea is to take the essence and discard the dregs from the two paradigms. To attain this, we propose a novel end-to-end one-stage framework with disentangling human-object detection and interaction classification in a cascade manner, namely Cascade Disentangling Network (CDN). The original intention of our framework is to keep the advantages of conventional one-stage methods, **directly and accurately locating the interactive human-object pairs**, and bring the advantages of two-stage methods into our one-stage framework, **disentangling human-object detection and interaction classification**. As shown in Figure 1(d), in our proposed framework, we design a human-object pair decoder based on the one-stage paradigm by removing the interaction classification function, namely HO-PD, and following an isolated interaction classifier. To instantiate our idea with an end-to-end manner, we design the HO-PD based on the previous state-of-the-art one-stage transformer-based HOI detector, HOI-Trans [39] and QPIC [28], where we remove the interaction classification head for each query and make it focus on human-object pairs detection. Otherwise, we design an independent HOI decoder for interaction classification to make the interaction classification unaffected by human-object detection. Therefore, there exists a core problem, *i.e.*, how to link the human-object pair and the corresponding action class. To address this problem, we initialize the query embedding of the HOI decoder with the output of the last layer of the HO-PD. In this case, the HOI decoder is able to learn the corresponding action category under the guidance of the query embedding and free out from the human-object detection task. Moreover, we design a decoupling dynamic re-weighting manner to handle the long-tailed problems in HOI detection.

Our contributions can be summarized threefold: (1) We conduct a detailed analysis of two conventional HOI detection paradigms, *i.e.*, two-stage and one-stage. (2) We propose a novel one-stage framework with a cascade disentangling decoder to combine the advantages of two-stage and one-stage methods. (3) Our method outperforms previous state-of-the-art methods by a large margin on the HOI detection task, especially achieves a $25.35\%$ performance gain on rare classes of HICO-Det.

## 2 Analysis of Two-stage and One-stage HOI detectors

We first introduce a unified formulation for the HOI detection problem. Given a human-centric image $\boldsymbol{I}$, the model $T(\cdot)$ is required to predict a set of HOI triplets $S = \{(b_i^h, b_i^o, a_i), i \in \{1, 2, \cdots, K\}\}$, where $b_i^h$, $b_i^o$ and $a_i$ denotes a human bounding-box, an object bounding-box and their corresponding action category, respectively.

**Two-stage HOI detector.** Two-stage detectors can be regarded as an instance-driven manner, detecting instances first and predicting interaction based on the detected instances. The two-stage detector divides $T(\cdot)$ into two stages, *i.e.*, detection $T_d(\cdot)$ and interaction classification $T_c(\cdot)$. In the first stage,

we suppose that $T_d(\cdot)$ produces $M$ human bounding-boxes and $N$ object bounding-boxes. Here the 'object' is a universal object which includes human as one class. Therefore, $T_d(\cdot)$ generates $M \times N$ human-object pairs. In general, the number of true-positive interactive human-object pairs, denoted as $K'$, is much smaller than $M \times N$. However, in the second stage, $T_c(\cdot)$ needs to scan all $M \times N$ pairs one by one and predict an action category with its corresponding confidence score. In this case, $T_c(\cdot)$ is required to inference $M \times N$ times to find $K'$ interactive pairs from $M \times N$ pairs. We argue that this manner causes three problems. Firstly, these models produce a more additional computational cost, whose time complexity is $\mathcal{O}(M \times N) \gg \mathcal{O}(K')$. Secondly, the imbalance between positive and negative samples makes the model easily overfit to negative samples. Thus the model tends to assign a 'no-interaction' class for human-object pairs with very high confidence, suppressing the true-positive samples. Thirdly, the accuracy of interaction classification is influenced by the non-end-to-end pipeline. Because the interaction classification is mostly based on the region features extracted by $T_d(\cdot)$, while the core of $T_d(\cdot)$ is to regress bounding-boxes and its extracted features pay more attention to the edge of regions, thereby such features are not good options for interaction classification, which needs more context. However, it is an excellent property for two-stage methods that disentangling detection and interaction classification makes each stage focus on its task and produce good results in each stage.

**One-stage HOI detector.** As for one-stage methods, they detect all HOI triplets $S$ directly and simultaneously with an end-to-end framework. Such paradigm has greatly relieved the three problems of two-stage methods, especially for efficiency, where the time complexity is reduced to $\mathcal{O}(K')$. Most one-stage methods are interaction-driven, which directly locate the interaction point [20] or interactive human-object pairs [39], thereby reducing negative sample interference. However, coupling human-object detection and interaction classification limit their performance because it is hard to generate a unified feature representation for two very different tasks. Though the parallel one-stage methods break HOI detection into two parallel branches, their interaction branch still suffers from multi-task learning. Specifically, the optimization target of interaction branch is $\mathcal{P}(e_h, e_o, a|\boldsymbol{V})$, where $e_h$ and $e_o$ are associative embeddings, *e.g.*, offset, to match interaction with human and object respectively. Therefore, even if detection is organized as an independent branch, the interaction branch must position humans and objects for the association. The set-based detectors couple detection and interaction completely, whose optimization function is $\mathcal{P}(b^h, b^o, a|\boldsymbol{V})$.

Next, we introduce a simple one-stage framework with disentangling human-object detection and interaction classification, namely CDN, to mine the benefits of two-stage and one-stage HOI detectors. Our CDN disentangles the original set-based one-stage optimization function into two cascade decoders. Firstly, we predict human-object pair by $\mathcal{P}(b^h, b^o|\boldsymbol{V})$. Secondly, we apply an isolated decoder to predict the action category by $\mathcal{P}(a|\boldsymbol{V}, b^h, b^o)$. More details are in the following.

## 3 Method

In this section, we will present a detailed introduction to the pipeline of our proposed CDN. In section 3.1, we present an overview of our framework and briefly introduce the pipeline. In section 3.2, we introduce the visual feature extractor. The cascade disentangling HOI decoder is introduced in section 3.3. Section 3.4 introduces a novel dynamic re-weighting mechanism that mitigates the long-tailed problem. The detailed training process and post-processing are discussed in section 3.5.

### 3.1 Overview

The architecture of our proposed CDN is illustrated in Figure 2. Our CDN is organized in a cascade manner with a visual feature extractor. Given an image, we first follow transformer-based detection methods [1, 39] to apply a CNN followed by a transformer encoder architecture to extract visual features into a sequence. Then we detect HOI triplets in two cascade decoders. Firstly, we apply the Human-Object Pair Decoder (HO-PD) to predict a set of human-object bounding-boxes pairs based on a set of learnable queries. Next, taking the output of the last layer of HO-PD as queries, an isolated interaction decoder is utilized to predict the action category for each query. Finally, the HOI triplets are formed by the output of the above two cascade decoders.

### 3.2 Visual Feature Extractor

We define the visual feature extractor by combining a CNN and a transformer encoder. Fed with an input image $\boldsymbol{I}$ with shape $(H, W, C)$, the CNN generates a feature map of shape $(H^{'}, W^{'}, D_b)$.

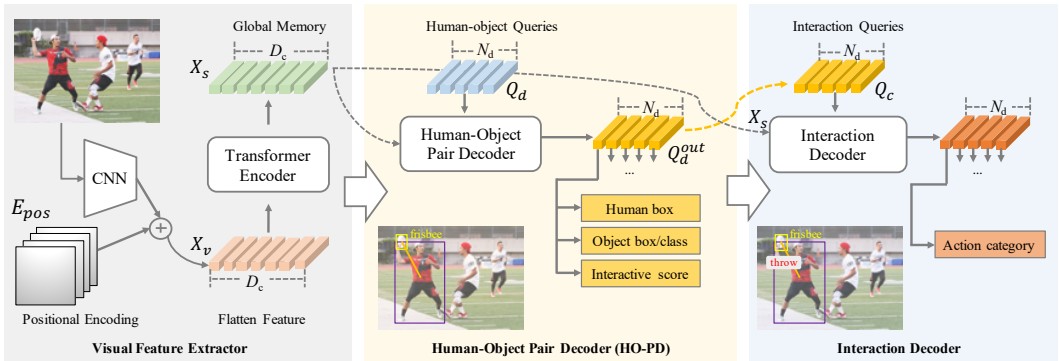

Figure 2: **The framework of our CDN**. It is comprised of three components:*Visual Feature Extractor*, *Human-Object Pair Decoder (HO-PD)* and *Interaction Decoder*. We first apply a CNN-transformer combined architecture to extract sequenced visual features $\boldsymbol{X}_s$. Then, we divide HOI detection into two cascade transformer-based decoders. Firstly, we regress the human-object bounding-box pairs based on $\boldsymbol{X}_s$ and a set of random-initialized queries $\boldsymbol{Q}_d$ by HO-PD. The interactive score is from a binary classification to determine whether the human-object pair is an interactive pair or not. Secondly, we predict one or many action categories for each predicted human-object pairs, where we take the output of HO-PD $\boldsymbol{Q}_d^{out}$ to initialize the interaction queries $\boldsymbol{Q}_c$ and aggregate information with $\boldsymbol{X}_s$. Finally, the HOI triplets are formed by the output of the cascade decoders.

Then, $D_b$ is reduced to $D_c$ by a projection convolution layer with a kernel size $1 \times 1$. Next, a flatten operator is used to generating the flatten feature $\boldsymbol{X}_v \in \mathcal{R}^{(H' \times W') \times D_c}$ by collapsing the spatial dimensions into one dimension. This flatten feature is then fed into a transformer encoder and the position encoding $\boldsymbol{E}_{pos} \in \mathcal{R}^{(H' \times W') \times D_c}$, which distinguishes the relative position in the sequence $\boldsymbol{X}_s \in \mathcal{R}^{(H' \times W') \times D_c}$. Thanks to the multi-head self-attention mechanism, the transformer encoder produces a feature map with richer contextual information by summarizing global information. The output of the encoder is denoted as global memory with a dimension of $D_c$.

### 3.3 Cascade Disentangling HOI Decoder

The cascade disentangling HOI decoder consists of two decoders: Human-Object Pair Decoder (HO-PD) and interaction decoder. Both decoders share the same architecture, a transformer-based decoder, with independent weights. In this subsection, we first introduce the general architecture of the decoder and then elaborate on the two decoders in detail, respectively.

**Transformer-based decoder**. We follow the transformer-based object detector DETR [1] to design the basic architecture in our cascade disentangling HOI decoder. We apply $N$ transformer decoder layers for each decoder and equip each decoder layer with several FFN heads for intermediate supervision. Specifically, each decoder layer is comprised of a self-attention module and a multi-head co-attention module. During feed-forward, fed into a set of learnable queries $\boldsymbol{Q} \in \mathcal{R}^{N_q \times C_q}$, each decoder layer first applies a self-attention module on all queries and then conducts a multi-head co-attention operation between queries and the sequenced visual features, and outputs a set of updated queries. For the FFN heads, each head is comprised of one or several MLP branches, and each branch is for a specific task, *e.g.*, regression, or classification. All queries share the same FFN heads. Therefore, each decoder can be simply represented as:

$$\boldsymbol{P} = f(\boldsymbol{Q}, \boldsymbol{X}_s, \boldsymbol{E}_{pos}). \tag{1}$$

Besides, the number of queries $N_q$ is determined by the number of positive samples of an image.

**HO-PD**. Firstly, we design the HO-PD to predict a set of human-object pairs from the sequenced visual features. To this end, we first randomly initialize a set of learnable queries $\boldsymbol{Q}_d \in \mathcal{R}^{N_d \times C_q}$ as HO queries. Then we apply a transformer-based decoder, which takes HO queries $\boldsymbol{Q}_d$ and sequenced visual features as input and applies three FFN heads for each query to predict human bounding-box, object bounding-box, and object class, which form a human-object pair. We also utilize an additional interactive score head to simply determine whether the human-object pair is an interactive pair or not by a binary classification. In this case, $\boldsymbol{P}$ is instantiated as $\boldsymbol{P}_{ho}$, which is consist of a set of

human-object pairs $\{(b_i^h, b_i^o), i \in \{1, 2, \cdots, N_d\}\}$. Thus, HO-PD can be denoted as:

$$\boldsymbol{P}_{ho} = f_d(\boldsymbol{Q}_d, \boldsymbol{X}_s, \boldsymbol{E}_{pos}). \tag{2}$$

In addition, we keep the output queries of the last layer of HO-PD as $\boldsymbol{Q}_d^{out}$ for the following step.

**Interaction Decoder.** Secondly, we propose the interaction decoder to classify the human-object queries and assign one or several action categories for each human-object query. To classify each human-object query one-to-one, we initialize $\boldsymbol{Q}_c$ with the output of HO-PD $\boldsymbol{Q}_d^{out}$. In this way, $\boldsymbol{Q}_d^{out}$ can provide prior knowledge to guide $\boldsymbol{Q}_c$ to learn the corresponding action categories for each human-object query. The other components and inputs are the same as HO-PD, which conducts self-attention among queries and co-attention with $\boldsymbol{X}_s$ and $\boldsymbol{E}_{pos}$. The final output is a set of action categories $\boldsymbol{P}_{cls} : \{a_i, i \in \{1, 2, \cdots, N_d\}\}$. Therefore, this process can be formulated as:

$$\boldsymbol{P}_{cls} = f_{cls}(\boldsymbol{Q}_d^{out}, \boldsymbol{X}_s, \boldsymbol{E}_{pos}). \tag{3}$$

In our proposed cascade disentangling HOI decoder, the task of HOI detection is disentangled into two relatively independent steps: human-object pairs detection and interaction classification. Therefore, each step can aggregate more related features to concentrate on its corresponding task.

### 3.4 Decoupling Dynamic Re-weighting

The HOI datasets usually have long-tail class distribution for both object class and action class. To alleviate the long-tail problem, we design a dynamic re-weighting mechanism for further improvements with a decoupling training strategy. In detail, we first train the whole model with regular losses. Then, we freeze the parameters of the visual feature extractor and only train the cascade disentangling decoders with a relatively small learning rate and the designed dynamic re-weighted losses.

During decoupling training, at each iteration, we apply two similar queues to accumulate number of each object class or action class. The queues are used as memory banks to accumulate training samples and truncate the accumulation with length $L_Q$ as sliding windows. In detail, $Q_o$ with length $L_Q^o$ to accumulate object number $N_i^o$ for each object class $i \in \{1, 2, \cdots, C_o\}$, and $Q_a$ with length $L_Q^a$ to accumulate interaction number $N_i^a$ for each action category $i \in \{1, 2, \cdots, C_a\}$. The dynamic re-weighting coefficients $w_{dynamic}$ are presented as follow:

$$w_i^a\big|_{i \in \{1,2,\cdots,C_a\}} = \Big(\frac{\sum_{i=1}^{C_a} N_i}{N_i}\Big)^{p_a}, \quad w_{bg}^a = \Big(\frac{\sum_{i=1}^{C_a} N_i}{N_{bg}^a}\Big)^{p_a}, \tag{4}$$

$$w_i^o\big|_{i \in \{1,2,\cdots,C_o\}} = \Big(\frac{\sum_{i=1}^{C_o} N_i}{N_i}\Big)^{p_o}, \quad w_{bg}^o = \Big(\frac{\sum_{i=1}^{C_o} N_i}{N_{bg}^o}\Big)^{p_o}, \tag{5}$$

where $N_i$ is the number of accumulated positive samples of category $i$ by the queues $Q_o$ and $Q_a$, $N_{bg}$ is the number of accumulated background samples, $C$ is the number of categories, and exponent $p$ is a hyper-parameter that adapts the magnitude of mitigation. Specifically, the weight of background class, $w_{bg}$, is designed to balance the positives and negatives. For the stability of the dynamic re-weighted training, the weight coefficients are initialized as $w_{static}$ with those calculated by 4 and 5 using the static number of object and action categories. The final dynamic weights are given as $w = \gamma w_{static} + (1 - \gamma) w_{dynamic}$, where $\gamma$ is a smooth factor, given as $min(0.999^{L_Q}, 0.9)$. The factor $\gamma$ transits $w$ from $w_{static}$ to $w_{dynamic}$ with the increasing of $L_Q$. Finally, the weights are used to the classification losses in a traditional way by multiplying each coefficient to each class and then calculating the summation.

### 3.5 Training and Post-processing

In this section, we introduce the training and inference processes in detail. Especially, we will introduce a novel Pair-wise Non-Maximal Suppression (PNMS) strategy in the inference process.

**Training.** Following the set-based training process of HOI-Trans [39] and QPIC [28], we first match each ground-truth with its best-matching prediction by the bipartite matching with the Hungarian algorithm. Then the loss is produced between the matched predictions and the corresponding ground truths for the final back-propagation. During matching, we consider the predictions of two cascade decoders together. The loss of CDN follows QPIC which is composed by five parts: the box regression

loss $L_b$, the intersection-over-union loss $L_{GIoU}$ [26], the interactive score loss $L_p$, the object class loss $L_c^o$, and the action category loss $L_c^a$. The target loss is the weighted sum of these parts as:

$$L = \sum_{k \in (h,o)} (\lambda_b L_b^k + \lambda_{GIoU} L_{GIoU}^k) + \lambda_p L_p + \lambda_o L_c^o + \lambda_a L_c^a, \qquad (6)$$

where $\lambda_b$, $\lambda_{GIoU}$, $\lambda_p$, $\lambda_o$ and $\lambda_a$ are the hyper-parameters for adjusting the weights of each loss.

**Inference.** The inference process is to composite the output of instance-related FFNs and the interaction-related FFN to form HOI triplets. By our cascade disentangling decoder architecture, the instance queries and the interaction queries are one-to-one corresponding, therefore, the five components <*human bounding box*, *object bounding box*, *object class*, *interactive score*, *action class*> can be homologous in each of the $N_d$ dimensions per FFN head. Formally, we generate the $i$-th output prediction as <$b_i^h$, $b_i^o$, $\mathrm{argmax}_k c_i^{hoi}(k)$>. The HOI triplet score $c_i^{hoi}$ is given by $c_i^{hoi} = c_i^a c_i^o c_i^p$, where $c_i^a$ and $c_i^o$ are the scores of interaction and object classification, respectively, and $c_i^p$ is the interactive score from the interactive FFN head for the query vector being an human-object pair.

**PNMS.** After sorting $c_i^{hoi}$ in descending order and generating the top $K$ HOI triplets, we design a pair-wise non-maximal suppression (PNMS) method to further filter out human-object pairs from pair-wise bounding boxes overlapping perspective. For two HOI triplets $m$ and $n$, the pair-wise overlap *PIoU* is calculated as:

$$PIoU(m,n) = \left( \frac{I(b_m^h, b_n^h)}{U(b_m^h, b_n^h)} \right)^\alpha \left( \frac{I(b_m^o, b_n^o)}{U(b_m^o, b_n^o)} \right)^\beta, \qquad (7)$$

where the operators $I$ and $U$ compute the intersection and union areas between the two boxes of $m$ and $n$, respectively. $\alpha$ and $\beta$ are the balancing parameters between humans and objects.

## 4 Experiments

In this section, we conduct comprehensive experiments to demonstrate the superiority of our designed CDN. In section 4.1, we briefly introduce the experimental benchmarks. Section 4.2 presents implementation details. Next, It is a detailed experimental comparison and analysis of two-stage and one-stage methods in section 4.3. In section 4.4, we compare our methods with the previous state-of-the-art methods. The ablation studies and components analysis are included in 4.5.

### 4.1 Datasets and Evaluation Metrics

**Datasets.** We carry out experiments on two widely-used HOI detection benchmarks: HICO-Det [2] and V-COCO [8]. We follow the standard evaluation scheme. HICO-Det consists of $47,776$ Creative Common images from Flickr ($38,118$ for training and $9,658$ for test) with more than $150K$ human-object pairs. It contains the same $80$ object categories as MS-COCO [21] and $117$ action categories. The objects and actions form $600$ classes of HOI triplets. V-COCO is derived from MS-COCO dataset, which consists of $5,400$ images in the trainval subset and $4,946$ images in the test subset. It has $29$ action categories ($25$ HOIs and $4$ body motions) and $80$ object categories. For both datasets, one person can interact with multiple objects in different ways at the same time.

**Evaluation Metrics.** Following the standard evaluation [2], we use the mean average precision (mAP) as the evaluation metric. For one positively predicted HOI triplet <human, object, action>, it needs to contain accurate human and object locations (box IoU with reference to GT box is greater than $0.5$) and correct object and action categories. Specifically, for HICO-Det, besides the full set of $600$ HOI classes, we also report the mAP over a rare set of $138$ HOI classes that have less than $10$ training instances and a non-rare set of the other $462$ HOI classes. For V-COCO, we report the role mAP for two scenarios: scenario 1 includes the cases even without any objects (for the four action categories of body motions), and scenario 2 ignores these cases.

### 4.2 Implementation Details

We implement three variant architectures of CDN: CDN-S, CDN-B, and CDN-L, where 'S', 'B', and 'L' denote small, base, and large, respectively. For CDN-S and CDN-B, we adopt ResNet-50 with a 6-layer transformer encoder as the visual feature extractor. For the cascade decoders, CDN-S is equipped with both 3-layer transformers, while CDN-B has a 6-layer transformer for each decoder. CDN-L only replaces the ResNet-50 with ResNet-101 in CDN-B. The reduced dimension size $D_c$

is set to 256. The number of queries $N_d$ is set to 64 for HICO-Det and 100 for V-COCO since the average number of positives for variant human-object pairs per image of HICO-Det is smaller than V-COCO. The human and object box FFNs have 3 linear layers with ReLU, while the object and action category FFNs have one linear layer. The code is provided in supplemental material.

During training, we initialize the network with the parameters of DETR [1] trained with the MS-COCO dataset. We set the weight coefficients $\lambda_b$, $\lambda_{GIoU}$, $\lambda_p$, $\lambda_o$ and $\lambda_a$ to 2.5, 1, 1, 1 and 1, respectively, which are exactly same with QPIC [28]. We optimize the network by AdamW [23] with the weight decay $10^{-4}$. We first train the whole model for 90 epochs with a learning rate of $10^{-4}$ decreased by 10 times at the 60th epoch. Then, during the decoupling training process, we fine-tune the cascade disentangling decoders together with the box, object, and action FFNs for 10 epochs with a learning rate of $10^{-5}$. We use both object and action dynamic re-weighting for HICO-Det and only action dynamic re-weighting for V-COCO. The re-weighting parameter $p$ is set to 0.7 for both object and action. The length $L_Q$ of training sample queue $Q$ for both object and action is set to $2 \times N_s$, where $N_s$ is the sample number of the training set. All experiments are conducted on the 8 Tesla V100 GPUs and CUDA10.2, with a batch size of 16.

For validation, we select 100 detection results with the highest scores and then adopt PNMS to further filter results. The threshold, $\alpha$, and $\beta$ of PNMS are set to 0.7, 1, and 0.5, respectively.

### 4.3 Experiment Analysis of Two-stage and One-stage Methods

In this part, we introduce a detailed experimental analysis of conventional two-stage and one-stage methods and our proposed CDN from the following three aspects.

**Human-object Pair Generation.** We first explore the quality of the human-object pairs generation between two-stage and one-stage methods. To attain this, we conduct a detailed experiment based on a representative two-stage method iCAN [6]. We first implement a PyTorch version iCAN as the baseline model, denoted as iCAN\*, which only applies human and object appearance with a COCO-pretrained Faster-RCNN detector [25]. For a fair comparison, we first fine-tune DETR on HICO-Det for 100 epochs only with the instance detection annotation based on COCO-pretrained weights. Then we combine the detected human and object bounding-boxes, whose confidences are greater than a threshold, one by one to generate human-object pairs denoted as iCAN† in Table 1. We train our CDN only with HO-PD for 100 epochs and get the human-object pairs from the output directly. Then, we graft the human-object pairs to the baseline model to extract box features and utilize the same interaction classifier in the second stage of iCAN\*. In this way, we degrade the number of pairs from $M \times N$ to $K'$, which means time complexity is reduced from $\mathcal{O}(M \times N)$ to $\mathcal{O}(K')$. Primarily, HO-PD significantly promotes mAP from 15.37 to 24.05, as shown in Table 1. This indicates that one-stage methods are much superior in human-object pair generation.

| Strategy | Full | Rare | Non-Rare |
|---|---|---|---|
| iCAN\* | 14.16 | 12.26 | 14.73 |
| iCAN † | 15.37 | 13.23 | 16.01 |
| HO-PD+iCAN\* | 24.05 | 18.32 | 25.76 |
| QPIC [28] | 29.07 | 21.85 | 31.23 |
| CDN-S base | **30.96** | **27.02** | **32.14** |

Table 1: **Analysis of Two-stage and One-stage Methods**. \* denotes our implemented PyTorch version iCAN [6] baseline model. † denotes replacing instance detection boxes given by a HICO-Det fine-tuned DETR detector to extract box features. 'HO-PD+iCAN\*' denotes replacing original one-by-one generated human-object pairs with our HO-PD generated. 'CDN-S base' denotes CDN-S w/o re-weighting and PNMS strategies.

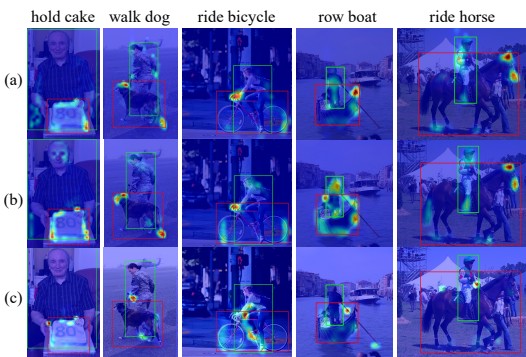

hold cake  walk dog  ride bicycle  row boat  ride horse

(a)

(b)

(c)

Figure 3: **Visualization of Feature Maps for Queries**. Visual attended features for query with top-1 score extracted from the last layer of the decoder of (a) QPIC, (b) HO-PD in CDN, and (c) interaction decoder in CDN. Zoom in for details.

**Interaction Classification.** We aim to study the interaction classification between conventional multi-task one-stage methods and our disentangled one-stage detector. We can regard QPIC [28] as a

| Method | Detector | Backbone | Extra | Default | | | Know Object | | |
|---|---|---|---|---|---|---|---|---|---|
| | | | | Full | Rare | Non-Rare | Full | Rare | Non-Rare |
| Two-stage Method: | | | | | | | | | |
| InteractNet [7] | COCO | ResNet-50-FPN | ✗ | 9.94 | 7.16 | 10.77 | - | - | - |
| GPNN [24] | COCO | Res-DCN-152 | ✗ | 13.11 | 9.34 | 14.23 | - | - | - |
| iCAN [6] | COCO | ResNet-50 | ✗ | 14.84 | 10.45 | 16.15 | 16.26 | 11.33 | 17.73 |
| No-Frills [9] | COCO | ResNet-152 | P | 17.18 | 12.17 | 18.68 | - | - | - |
| PMFNet [30] | COCO | ResNet-50-FPN | P | 17.46 | 15.65 | 18.00 | 20.34 | 17.47 | 21.20 |
| CHGNet [31] | COCO | ResNet-50 | ✗ | 17.57 | 16.85 | 17.78 | 21.00 | 20.74 | 21.08 |
| DRG [5] | COCO | ResNet-50-FPN | T | 19.26 | 17.74 | 19.71 | 23.40 | 21.75 | 23.89 |
| VCL [12] | COCO | ResNet-50 | ✗ | 19.43 | 16.55 | 20.29 | 22.00 | 19.09 | 22.87 |
| IP-Net [32] | COCO | Hourglass-104 | ✗ | 19.56 | 12.79 | 21.58 | 22.05 | 15.77 | 23.92 |
| VSGNet [29] | COCO | ResNet-152 | ✗ | 19.80 | 16.05 | 20.91 | - | - | - |
| FCMNet [22] | COCO | ResNet-50 | ✗ | 20.41 | 17.34 | 21.56 | 22.04 | 18.97 | 23.12 |
| ACP [15] | COCO | ResNet-152 | T | 20.59 | 15.92 | 21.98 | - | - | - |
| PD-Net [35] | COCO | ResNet-152-FPN | T | 20.81 | 15.90 | 22.28 | 24.78 | 18.88 | 26.54 |
| DJ-RN [16] | COCO | ResNet-50 | P | 21.34 | 18.53 | 22.18 | 23.69 | 20.64 | 24.60 |
| IDN [17] | COCO | ResNet-50 | ✗ | 23.36 | 22.47 | 23.63 | 26.43 | 25.01 | 26.85 |
| One-stage Method: | | | | | | | | | |
| UnionDet [13] | COCO | ResNet-50-FPN | ✗ | 17.58 | 11.72 | 19.33 | 19.76 | 14.68 | 21.27 |
| DIRV [4] | COCO | EfficientDet-d3 | ✗ | 21.78 | 16.38 | 23.39 | 25.52 | 20.84 | 26.92 |
| PPDM-Hourglass [20] | HICO-DET | Hourglass-104 | ✗ | 21.94 | 13.97 | 24.32 | 24.81 | 17.09 | 27.12 |
| HOI-Trans [39] | HICO-DET | ResNet-50 | ✗ | 23.46 | 16.91 | 25.41 | 26.15 | 19.24 | 28.22 |
| GG-Net [37] | HICO-DET | Hourglass-104 | ✗ | 23.47 | 16.48 | 25.60 | 27.36 | 20.23 | 29.48 |
| ATL [11] | HICO-DET | ResNet-50 | ✗ | 23.81 | 17.43 | 25.72 | 27.38 | 22.09 | 28.96 |
| HOTR [14] | HICO-DET | ResNet-50 | ✗ | 25.10 | 17.34 | 27.42 | - | - | - |
| AS-Net [3] | HICO-DET | ResNet-50 | ✗ | 28.87 | 24.25 | 30.25 | 31.74 | 27.07 | 33.14 |
| QPIC [28] | HICO-DET | ResNet-50 | ✗ | 29.07 | 21.85 | 31.23 | 31.68 | 24.14 | 33.93 |
| CDN-S | HICO-DET | ResNet-50 | ✗ | 31.44 | 27.39 | 32.64 | 34.09 | 29.63 | 35.42 |
| CDN-B | HICO-DET | ResNet-50 | ✗ | 31.78 | **27.55** | 33.05 | 34.53 | **29.73** | 35.96 |
| CDN-L | HICO-DET | ResNet-101 | ✗ | **32.07** | 27.19 | **33.53** | **34.79** | 29.48 | **36.38** |

Table 2: **Performance comparison on the HICO-Det test set.** The 'P', 'T' represent human pose information and the language feature, respectively.

multi-task version of our CDN. Table 1 shows that our 'CDN-S base' (w/o re-weighting and PNMS strategies) has achieved mAP 30.96 with $6.50\%$ relative mAP gain compared to QPIC. Especially, our 'CDN-S base' significantly outperforms QPIC for rare classes with a $23.66\%$ improvement. The performance of rare classes can partly reflect the accuracy of interaction classification.

**Feature Learning.** This part discusses the differences in feature learning between the conventional one-stage method, QPIC, and our CDN from a qualitative view. As shown in Figure 3, we visualized the feature maps extracted from the last layer of the decoder of QPIC, the HO-PD, and the interaction decoder in our CDN. We can see that HO-PD and QPIC attend very similar regions, *e.g.,* the boundaries of humans and objects and the human-object contact areas, which are beneficial for locating the interactive human-object pairs. However, the interaction decoder concentrates on human-pose and the regions that contribute to understanding human actions. As for the specific case, for example, for 'hold cake', HO-PD in CDN attends to the boundaries of the cake while the interaction decoder in CDN concentrates on the interaction context, i.e., the human's hands holding the cake. Thus, it shows that CDN disentangles the human-object detection and interaction classification. For 'ride horse', HO-PD in CDN emphasizes the overall feature of the human and the horse, and the highlight parts are the edges of the human and horse. For the interaction decoder in CDN, the highlighted part emphasizes the interaction context, i.e., the human carries the rope when riding a horse. Finally, QPIC somehow combines the two highlights, but both are not obvious.

## 4.4 Comparison to State-of-the-Art

We conduct experiments on HICO-Det and V-COCO benchmarks to verify the effectiveness of our proposed methods. For HICO-Det dataset as shown in Table 2, comparing to the previous state-of-the-art two-stage method FCMNet [22] with ResNet-50 as backbone, our CDN-B significantly promotes mAP from $20.41$ to $31.78$, with a relative gain of $55.71\%$. Even compared with PD-Net [36] which adopts extra language feature and DJ-RN [16] which utilizes extra human pose features, CDN-B achieves $52.71\%$ and $48.92\%$ relative mAP gains, respectively. When comparing to the one-stage method AS-Net [3] and QPIC [28] which also adopt transformer-based detector architecture, CDN-B attains $10.08\%$ and $9.32\%$ point relative mAP gains, respectively. Table 3 shows the comparisons on V-COCO dataset. CDN-B achieves $62.29$ $AP_{role}$ on Scenario 1 and $64.42$ $AP_{role}$ on Scenario 2, which significantly outperform previous state-of-the-art method QPIC with relative $5.94\%$ and $5.61\%$

| Method | Extra | $\text{AP}^{S1}_{role}$ | $\text{AP}^{S2}_{role}$ |
|---|---|---|---|
| Two-stage Method: | | | |
| InteractNet [7] | ✗ | 40.0 | - |
| GPNN [24] | ✗ | 44.0 | - |
| iCAN [6] | ✗ | 45.3 | 52.4 |
| TIN [18] | ✗ | 47.8 | 54.2 |
| VCL [12] | ✗ | 48.3 | - |
| DRG [5] | T | 51.0 | - |
| IP-Net [32] | ✗ | 51.0 | - |
| VSGNet [29] | ✗ | 51.8 | 57.0 |
| PMFNet [30] | P | 52.0 | - |
| PD-Net [35] | T | 52.6 | - |
| CHGNet [31] | ✗ | 52.7 | - |
| FCMNet [22] | ✗ | 53.1 | - |
| ACP [15] | T | 53.23 | - |
| IDN [17] | ✗ | 53.3 | 60.3 |
| One-stage Method: | | | |
| UnionDet [13] | ✗ | 47.5 | 56.2 |
| HOI-Trans [39] | ✗ | 52.9 | - |
| AS-Net [3] | ✗ | 53.9 | - |
| GG-Net [37] | ✗ | 54.7 | - |
| HOTR [14] | ✗ | 55.2 | 64.4 |
| DIRV [4] | ✗ | 56.1 | - |
| QPIC [28] | ✗ | 58.8 | 61.0 |
| CDN-S | ✗ | 61.68 | 63.77 |
| CDN-B | ✗ | 62.29 | 64.42 |
| CDN-L | ✗ | **63.91** | **65.89** |

Table 3: **Performance comparison on the V-COCO test set**. The 'P', 'T' represent the human pose information and the language feature, respectively.

| Strategy | Full | Rare | Non-Rare |
|---|---|---|---|
| QPIC [28] | 29.07 | 21.85 | 31.23 |
| *base* | 31.06 | 26.68 | 32.36 |
| *+ re-weighting* | 31.38 | 27.36 | 32.58 |
| *+ PNMS* | **31.78** | **27.55** | **33.05** |

(a) **Strategies:** Analysis of improvements by various training strategies.

| Strategy | $L_Q$ | p | Full | Rare | Non-Rare |
|---|---|---|---|---|---|
| *base* | - | - | 31.06 | 26.68 | 32.36 |
| *decouple* | - | - | 30.90 | 26.09 | 32.33 |
| *static* | - | 0.7 | 31.25 | 27.12 | 32.49 |
| *dynamic* | $2 \times N_s$ | 0.8 | 31.33 | 27.45 | 32.49 |
| *dynamic* | $1 \times N_s$ | 0.7 | 31.34 | **27.48** | 32.49 |
| *dynamic* | $2 \times N_s$ | 0.7 | **31.38** | 27.36 | **32.58** |

(b) **Dynamic re-weighting:** Analysis of decouple training with dynamic re-weighted losses, *i.e.*, different queue length $L_Q$, coefficient $p$ and dynamic or static.

| $\alpha$ | $\beta$ | *thres* | Full | Rare | Non-Rare |
|---|---|---|---|---|---|
| - | - | - | 31.38 | 27.36 | 32.58 |
| *1* | *1* | *0.8* | 31.66 | 27.46 | 32.91 |
| *1* | *1* | *0.7* | 31.75 | 27.50 | 33.03 |
| *1* | *0.7* | *0.7* | 31.77 | 27.54 | 33.03 |
| *1* | *0.5* | *0.8* | 31.75 | 27.51 | 33.02 |
| *1* | *0.5* | *0.7* | **31.78** | **27.55** | **33.05** |

(c) **PNMS:** The effects of different settings of PNMS coefficients, *i.e.*, $\alpha$, $\beta$, and *thres* denotes threshold.

Table 4: **Ablation studies of our proposed method on the HICO-Det test set.** We carry out all experiments based on the base model (CDN-B).

gains, respectively. As for efficiency analysis, CDN-S has almost the same number of parameters and flops compared to QPIC, but CDN-S achieves mAP 31.44 on HICO-Det, $8.15\%$ higher than QPIC.

## 4.5 Ablation Study

In this subsection, we analyse the effectiveness of the proposed strategies and components in detail. All experiments are eventuated on the HICO-Det dataset. The performance of each strategy is evaluated in Table 4a. The five hyper-parameters of the training loss in 7 follow QPIC [28]. The ablation study of the two hyper-parameters in the re-weighting is shown in Table 4b, and that of the three hyper-parameters in the PNMS is shown in Table 4c. We carry out all experiments based on the model CDN-B with ResNet-50 as backbone.

**Strategies.** As shown Table 4a, our pure model without any additional post-processing operation, namely base model, achieves mAP 31.06, promoting 1.99 compared with QPIC [28]. Especially, the base model significantly promotes mAP for rare classes from 21.85 to 26.68 compared to QPIC. It indicates the superiority of the architecture of disentangling human-object detection and interaction classification. The re-weighted training further promotes mAP to 31.38, with a gain of 0.32, and the gain mainly lies in rare classes. Finally, the PNMS further improves mAP to 31.78.

**Dynamic re-weighting.** In this part, we conduct experiments to evaluate the components in the dynamic re-weighted training strategy based on the base model as shown in Table 4b. If we only decouple training without re-weighting, the model achieves mAP 30.90, which is lower than the base model. Therefore, it shows that the performance gain does not come from a longer training process. Adding static weights $w_{static}$ to losses promotes mAP to 31.25. The dynamic re-weighting method improves the re-weighting effect since it captures the real-time weight of each class for each real-time sample during training. Thus it can sufficiently dig information from every single sample to achieve the best overall performance. Our method obtains best result mAP 31.38 when $L_Q = 2 \times N_s$ and $p = 0.7$.

**PNMS.** On the basis of the model after re-weighted training, we compare the variance by different parameter settings of the PNMS strategy, which is shown in Table 4c. We fix the human box balance

factor $\alpha$ to 1. Then we tune the object box balance factor $\beta$ and the threshold of the *PIoU* to filter pair-wise boxes. We achieve best performance mAP 31.78 when $\beta = 0.5$ and *thres* = 0.7. The fact that $\beta$ is smaller than $\alpha$, indicates that the overall performance is more sensitive to human boxes compared with object boxes in our framework.

## 5   Related Work

**Two-stage Methods.** Most previous HOI detectors are with a two-stage paradigm [6, 2, 18, 5]. Firstly, a fine-tuned detector [25, 10] is applied to detect the instances. Secondly, generating the human-object pairs by matching detected human and object one by one, and then feeding them into an interaction classifier. To improve the interaction classification, some extra features were applied, such as human pose [27, 19, 9], human parts [38, 30, 16], and language features [33, 5, 22, 15]. Besides, some two-stage methods [24, 29, 31, 34, 38] applied graph neural networks to model the interactions.

**One-stage Methods**. One-stage methods detect HOI triplets directly [20, 32, 13, 39, 3, 28, 14]. In detail, [20, 32] proposed a point-based interaction detection method which performs inference at each interaction key point. [13] proposed an anchor-based method to predict the interactions for each human-object union box. Recently, set-based detection approach has been proposed to handle HOI detection as a set prediction problem [39, 3, 28]. Specifically, [39, 28] designed a transformer encoder-decoder architecture to directly predict HOI detection results in an end-to-end manner, while [3] utilized parallel instance and interaction decoder branches to adaptively aggregate the HOI triplets.

## 6   Conclusion

In this paper, we explore the essential pros and cons of two-stage and one-stage HOI detection in detail. We propose a novel one-stage framework with disentangling human-object detection and interaction classification in a cascade manner. Our CDN can keep the advantage of one-stage methods, directly and accurately locating the interactive human-object pairs, and bring the benefit of two-stage methods, disentangling detection and interaction classification. Our novel paradigm has outperformed previous methods by margins. However, we only implement a specific version to mine the benefits of two-stage and one-stage methods. In the future, we plan to apply our idea with more general one-stage methods and introduce more advantages of two-stage methods into the one-stage framework.

## Potential Negative Societal Impacts

Similar to many other AI technologies, our proposed CDN itself is harmless. However, someone might utilize it for military purposes or apply it to any other malicious human activities detection, which might negatively impact society. Therefore, we encourage well-intentioned consideration before adopting our technique.

## Acknowledgments and Disclosure of Funding

This research is partly supported by National Natural Science Foundation of China (Grant 61876177), Beijing Natural Science Foundation (4202034), Fundamental Research Funds for the Central Universities, Zhejiang Lab (No. 2019KD0AB04).

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
