# OpenReview forum: "Mining the Benefits of Two-stage and  One-stage HOI Detection"
_NeurIPS.cc/2021/Conference — NeurIPS 2021 Poster_

### Official Review · Reviewer_LC3p · 2021-07-09

**Rating:** 6
**Confidence:** 5

**Summary:**

This paper explores the difference between the one- and two-stage based HOI detection methods and a corresponding method to conduct the human-object interactive pair detection and interaction classification separately, based on the Transformer structure and previous work QPIC. Moreover, two distinct methods named dynamic re-weighting and PNMS are also proposed. In experiments, the proposed model achieves impressive performance and outperforms its predecessor, QPIC.

Even the performance is good, comparing to QPIC, the proposed method is limited in design novelty. Moreover, some experiment details, especially the core separated pair detection and interaction classification performance contribution, are not verified.

Overall, I think this work is interesting but still has a gap to achieve the acceptance bar of NeurIPS. Detailed comments are given below, and looking forward to the authors' response.

**Limitations And Societal Impact:**

Yes.
Method and experiment limitations are listed above.

**Main Review:**

1. Originality:
The proposed model is basically based on QPIC. The two decoders are different. However, the performance gap between the two methods is small if removing the post-processing and re-weighting from the experiments. Thus, I think the novelty is marginal.

2. Quality:
The implementation and experiments are reasonably good. The discussion about the one- and two-stage methods are clear and insightful.

3. Clarity:
The model part is good, but some experiment details, especially the contributions of hyperparameters and strategies, are not clear. This makes me concern about the actual performance contribution of the core novelty, i.e., the separated pair detection and interaction classification. The code is given.

4. Significance:
This paper has given some good points and analysis in terms of the discussion about the one- and two- stages.
In terms of the model design, the novelty is not so significant to afford the claim.
Regarding the experiment, I think some necessary tests are missing, e.g., the detailed interactive pair detection quality comparison, which is the core contribution. Simple HOI mAP cannot give enough hints and cues.

Detailed comments:

Pros:
+ Good discussion about the difference between one-stage and two-stage methods. This is useful for the HOI community.

+ Code and method description make the readers easy to follow.

+ Impressive performance.

Cons:
- Though the proposed method is indeed different from previous methods, but is still a kind of two-stage in my opinion. In L277-279, the authors think the proposed method is one-stage. Nevertheless, from the previous discussion, I think CDN is more like a two-stage method. The difference is depended on the h-o pairing process. For example, TIN [17] also has the neg pair filtering process (NIS). However, this point does not affect the contribution.
- In training, the two decoders are trained together. So this makes the claim unaffected (L59) confusing and may cause misleading. Moreover, I think it would also make the disentangling claim in the intro weird. I suggest the authors better clarify this. Because after reading the intro, readers may think the h-o pair detection and interaction classification are *totally* separated in both training and inference. This is important and also related to the above naming of one or two stages. I think the authors think the difference lies at the inference stage.
- L79: M*N, this is, in fact, not very accurate. In HICO-DET, human-human interactions are also considered. Thus, the pairs can be M*(M+N).  This should be clarified in both method explanation and implementation.
- The interaction score looks weird. This is not mentioned in the intro but actually may play an important role in training and inference. First, what is the difference between it and interactiveness defined in TIN [17]? They seem to be the same and should be discussed. Moreover, how the supervision of the binary classification (a pair is interactive or non-interactive) affect the performance? Do the binary labels convert from the HOI labels?
- L88-91: the discussion about the feature of one-stage is interesting, but from current results, if SOTA two-stage methods adopt the detected boxes from the finetuned detector (on HICO-DET) or transformer, their performances are pretty close with recent one-stage methods. So what kind of feature is suitable for the interaction understanding? Please give some insightful discussion or give a small test on this.
- L170, "each step can aggregate more related features to concentrate on its corresponding task", this is also confusing because the two decoders are trained together, and their gradients are entangled. Given the black-box property, I think this needs more verification instead of the good performance only.
- The Decoupling Dynamic Re-weighting part is similar to the method of Frustratingly Simple Few-Shot Object Detection. Please discuss this.
- Sec. 3.4 and 3.5 look not significantly related to the main contribution. Moreover, their performance contributions are not very clear. Does the 30.90 mAP come from the -B or -L setting? If using Res-50, compared to the QPIC (29.07), the gap is about one mAP. However, if using Res-101, the gap is less, as the QPIC with res-101 is about 29.90 [27] (missing in Tab.2). From the ablation study, the post-processing and re-weighting affect the performance a lot. So how the exact performance improvement of the core disentangling contribution? Please clarify this.
- About nine hyperparameters, how would they affect the performance? The ablation study does not give all analyses. Moreover, how they are decided. In Tab. 4, are they tuned based on the test set?
- L213 and 187, two alphas. L187 is p?
- Tab.1: the h-o pair quality can be directly compared by the interaction binary classification and box IoU.
- Fig 3: the difference is not so evident for me. Please give some more discussion according to the figs? For example, for ride-horse, does QPIC attention look better?

--Post-review: Appreciate the response from the authors. After reading the reviews and responses, my most concerns were addressed. I raising the rating to 6. For more details, please refer to the below response.

**Time Spent Reviewing:**

8

---

> ### Author Response · Authors · 2021-08-10
> **The Response to Reviewer LC3p**
>
> We sincerely thank you for your effort and valuable comments.
>
> ---
>
> **Q1**: Originality: The performance gap … novelty is marginal.
>
> **A1**: Without post-processing and re-weighting modules, our base model also achieves a significate performance gain. Specifically, our CDN-S (ResNet-50 w.t. 3+3 decoder layers, removing extra process) has also achieved 30.96 mAP, outperforming 29.07 mAP achieved by QPIC (ResNet-50) by 1.89 mAP in the HICO-Det dataset.
>
> Therefore, we argue that this gain can well verify the effectiveness of our core idea, i.e., mining the two-stage and one-stage HOI detections. Furthermore, the other reviewers all confirm that our performance is "superior", "notable" and "significant boosts".
>
> ---
>
> **Q2**: Clarity: Some experiment details … not clear.
>
> **A2**: The five hyper-parameters of the training loss in Eq. (5) follow the same settings in QPIC for a fair comparison.
>
> The tunable strategies and parameters in re-weighting and PNMS modules are shown in Table 4 as ablation studies.
>
> ---
>
> **Q3**: Significance: Some necessary tests … enough hints and cues.
>
> **A3**: For iCAN+, QPIC and CDN-S, we merge the action classes to binary class for ground-truths and predictions and remove the duplicated <h, o> pair. The mAPs are 20.39, 36.34, and 37.61, respectively. The results show that the <h, o> pair detection quality of HO-PD is convincing.
>
> ---
>
> **Q4**: Though the proposed method … not affect the contribution.
>
> **A4**: In our opinion, two-stage methods follow the detect and pair routine, i.e., applying an additional instance detector to detect human and object instances in the first stage and then pairing the human and object in the second stage.
>
> On the contrary, one-stage methods directly predict HOI pairs in an end-to-end framework. Our proposed method is somehow a combination of two-stage and one-stage HOI detection methods, but we prefer it as one-stage.
>
> We use cascade decoders to disentangle detection and interaction classification. From this perspective, it is like a two-stage method. However, as mentioned before, the key point of the one-stage method is to directly detect human-object pairs instead of detecting bounding boxes of objects and then pairing them.
>
> Therefore, the proposed method keeps the main characteristic of the one-stage method. Thus we attribute it to a one-stage method.
>
> ---
>
> **Q5**: In training, the two decoders … lies at the inference stage.
>
> **A5**: Thank you for the suggestion. We will change the “unaffected” to “as unaffected as possible” and refine our introduction section to clarify the meaning of disentangling better to avoid misunderstanding. In our revised version, we will clearly claim that the “h-o pair detection” decoder and the “interaction classification” decoder are trained together in an end-to-end manner, which means they are not totally separated during training. In fact, we adopt “disentangling” to indicate that two decoders predict different results (h-o pair detection and interaction classification) respectively in a cascade manner, which is quite different from previous works that predict them simultaneously. The way we use the expression of “disentangling” is common in the community (e.g., [1][2]). We believe that with additional descriptions according to your suggestions, readers will not be confused here.
>
> ---
>
> **Q6**: L79: MN, … implementation.
>
> **A6**: The “object” is a universal object, which includes humans as one class. Therefore, “N” includes human detections, and the number of human-object pairs is MN instead of M(M+N). We will clarify this point in the paper.
>
> ---
>
> **Q7**: The interaction score … from the HOI labels?
>
> **A7**: TIN is an insightful work to perform non-interaction suppression through learning an interactive score with an interactiveness network in a two-stage framework. In our work, the interaction score is from a binary classification to determine whether the human-object pair is an interactive pair or not. The HOI labels convert the binary labels. It has a limited effect on performance in our experiments, and the performance variations are less than 0.06.
>
> ---
>
> **Q8**: L88-91: the discussion … test on this.
>
> **A8**: We are confused by the description "SOTA one-stage methods ... close with recent one-stage methods." We guess that the "SOTA one-stage methods" could refer to the "SOTA two-stage methods."
>
> We agree with the reviewer that the two-stage methods benefit from the fine-tuned detector. The human-object detectors fine-tuned on the HICO-Det detect fewer non-interactive instances since HICO-Det is only annotated with interactive human-object bounding-boxes. However, we argue that it is essential for the two-stage methods to train the interaction classification jointly with the feature extractors (backbone and FPN) of instances detectors (e.g., very recent method SCG[3] shows a 1.2% performance gain from end-to-end training). Therefore, we consider that it is sub-optimal to apply human and object region features extracted from bounding-boxes detectors directly to classify interaction. And fine-tuning the feature extractor end-to-end is more suitable for the interaction understanding.
>
> As claimed in L88-91, the context feature from the whole image is important for interaction understanding. In this way, the model can extract the most suitable features from the image with a direct interaction loss supervision instead of the fixed local region features for the object detection task. Moreover, the transformer-based network is a pretty well feature extractor for interaction understanding. Specifically, it could provide a long-range receptive field and make the interaction query integrate different features for different interactions through a cross-attention operation and set matching loss.
>
> Additionally, the hand-designed features, e.g., human posture and linguistic features, can also benefit interaction understanding, but effectively integrating such information into a simple end-to-end framework is also a problem worth studying.
>
> ---
>
> **Q9**: L170, "each step can aggregate … the good performance only.
>
> **A9**: Though the gradients are entangled, we apply different objectives and losses to supervise the two decoders to ensure they learn different features for their own tasks. Specifically, in Eq.(5), $L_b^k$, $L_{GIoU}^k$, $L_p$, and $L_c^o$ supervise the first cascade decoder (HO-PD), while $L_c^a$ supervises the second cascade decoder (Interaction Decoder).
>
> We also give some visualizations in Fig. 3, where features in two decoders are shown. It illustrates the different focuses of the two decoders. Specifically, the HO-PD focuses more on the boundaries of humans and objects, while the interaction decoder concentrates more on interaction context, such as human postures.
>
> ---
>
> **Q10**: The Decoupling … discuss this.
>
> **A10**: The mentioned method finetunes the last layer of the trained detector on a smaller balanced dataset to promote the performance of rare classes. In our method, the “decouple” setting is similar to only finetune the two cascade decoders. Meanwhile, our training is implemented on the original training set. Then, our key contribution is to accumulate the number of samples during the whole training process for each class of both the objects and the actions and then calculates the weight for each class by Eq. (4). A queue truncates the accumulation with length $L_Q$ as a sliding window. Then the weights are used to re-weight the training for the object and action classification. As we claim in L314-316, the method captures the real-time weight of each class for each real-time sample during training. Thus, it digs information sufficiently for every single sample.
>
> ---
>
> **Q11**: Sec. 3.4 and 3.5 … Please clarify this.
>
> **A11**: The performance contributions of re-weighting (Sec. 3.4) and PNMS (Sec. 3.5) are shown in Table 4 (a). The “base” setting without re-weighting and PNMS indicates the core disentangling contribution, and it achieves 31.06 mAP (not 30.90 mAP), which outperforms 1.99 mAP compared to QPIC (29.07). The re-weighting and PNMS further promote 0.72 mAP in total. All the ablation study in Table 4 comes from the CDN-B setting (ResNet-50 as the backbone).
>
> ---
>
> **Q12**: About nine hyperparameters … the test set?
>
> **A12**: The five hyper-parameters of the training loss in Eq. (5) follow QPIC. The ablation study of the two hyper-parameters in the re-weighting is shown in Table 4 (b), and the three hyper-parameters in the PNMS is shown in Table 4 (c). We did not deliberately tune these five parameters and only give an ablation analysis. Some analysis of the parameters is given in L321-323.
>
> ---
>
> **Q13**: L213 and 187, two alphas. L187 is p?
>
> **A13**: Thanks for pointing this out. Here the symbol in L187 should be a new symbol like gamma (not p in Eq. 4), and gamma is a smooth factor.
>
> ---
>
> **Q14**: Fig 3: the difference … look better?
>
> **A14**: We give an overall discussion in L 282-288. As for the specific case, for example, for hold cake, HO-PD in CDN (Fig. 3 b) attends to the boundaries of the book. At the same time, the interaction decoder in CDN (Fig. 3 c) concentrates on the interaction context, i.e., the human’s hand holding the cake. Thus, it shows that the CDN well disentangle the human-object detection and interaction classification.
>
> For ride-horses, HO-PD in CDN (Fig. 3 b) emphasizes the overall feature of the human and the horse, and the highlighted parts are the edges of the human and horse. For the interaction decoder in CDN (Fig. 3 c), the highlighted parts emphasize the interaction context, i.e., the human carries the rope when riding a horse. Finally, QPIC (Fig. 3 a) somehow combines the two highlights, but both are not obvious.
>
> ---
>
> [1] Revisiting the sibling head in object detector. In CVPR 2020.
>
> [2] Disentangle Your Dense Object Detector. arXiv 2021.
>
> [3] Spatially Conditioned Graphs for Detecting Human-Object Interactions. arXiv 2021.

---

> > ### Comment · Reviewer_LC3p · 2021-08-18
> > **My concerns are mostly addressed.**
> >
> > After reading the response and the other reviews, my opinions are listed as follows:
> >
> > - One- or two- stage: the authors have discussed their definition, i.e., "Our proposed method is somehow a combination of two-stage and one-stage HOI detection methods, but we prefer it as one-stage." I thought the one- or two- stage definition is somehow ambiguous, and the authors also clarified their method lies at the middle place between one- and two- but is more like one-stage. This is according to the precondition. Overall, if the authors can add more discussions in the next version and revised the corresponding content to provide a more clear claim, I think it is fairly enough to address my concern.
> >
> > - The claim about the joint training of two decoders: the authors have responded that they will revise this part, addressed.
> >
> > - The performance improvement: the authors have clarified the comparison details and my concern is addressed.
> >
> > - Model details like hyperparameters: clarified and addressed.
> >
> > - Binary interaction score: This setting is similar to the interactiveness label generation (transferred from the HOI labels) and binary classification setting [17], thus, the next version needs a discussion to avoid ambiguity and clarification about its performance effect.
> >
> > - Supplementary test of binary classification: clearly addressed.
> >
> > - MN: addressed after adding the clarification.
> >
> > - Discussion about the backbone: interesting and useful, hope it can be added to the next version.
> >
> > - Entangled gradient: addressed.
> >
> > - Sec. 3.4 and 3.5: addressed. More clarification and revision are essential to give more details to readers.
> >
> > - alpha, p: clarified.
> >
> > - Comparison with QPIC: addressed.
> >
> > - Fig 3: I still think a more clear and vivid visualized indicator would be better. The liter explanation cannot fully and clearly point out the differences.
> >
> > Overall, I appreciate the responses from the authors and most of my concerns are addressed. However, some of them still remain, e.g., the clear definition and claim about the one- or two- stage, somehow unclear performance contribution of several sub-modules. So I think a major revision is essential to provide more clear illustrations and clarifications. I'm raising the rating from 5 to 6.

---

> > > ### Author Response · Authors · 2021-08-30
> > > **The Response to Reviewer LC3p**
> > >
> > > Thanks for your response and appreciation. We will update all the contents of our rebuttals and provide more representative visualization results to the revised version. Specifically, we will give detailed definitions about the one- or two-stage methods and provide a clearer claim about our proposed method. Then, we will discuss the setting of the interactive score and its effect on the performance. We will also give some discussion about the backbone and clearer descriptions of sec. 3.4 and 3.5 in the revised version. For fig. 3, we will point out the differences between different methods in detail. We will also clearly illustrate the performance contribution of submodules. All the other detailed issues in the rebuttals will be updated in the final version. Thank you again for your effort and valuable suggestions.

---

### Official Review · Reviewer_ez6V · 2021-07-12

**Rating:** 8
**Confidence:** 4

**Summary:**

Inspired by its success on various vision and NLP tasks, Transformer based architectures have recently been adopted and achieved impressive performance on the task of HOI detection [3, 27, 38]. This paper builds upon these work and further innovates on the decoder architecture. The idea is that instead of having a single stage decoder which generates human and object bounding boxes and object and interaction class labels altogether, the decoding process is divided into two stages, where a first decoder is used to generate interactive human-object pairs, followed by a second decoder that predicts the interaction class. This resembles the traditional two stage HOI detectors which first detect individual objects followed by classifying the category of each pair. However, thanks to the development of Transformer based methods [3, 27, 28], one can directly predict human-object pairs in one go rather than relying on individual object detection and pairing. Meanwhile, the proposed framework can still build upon the same intuition which is to decouple interaction detection and classification so the network can just focus on one task at once. Besides this main architectural change, the proposed approach also features some new techniques on training (Sec. 3.4 "dynamic re-weighting") and post processing (Sec. 3.5 "pair-wise non-maximal suppression"). The approach is evaluated on standard datasets (HICO-DET and V-COCO) and has achieved notable gains over SOTAs in detection mAP.

**Limitations And Societal Impact:**

Yes, the authors have discussed the potential impact of their work in a dedicated section after the conclusion.

**Main Review:**

### Strengths
- The proposed approach is novel and sound. The two stage pipeline (i.e. proposal - classification) has already shown wide success in the object detection domain (e.g. Faster R-CNN). Many two stage approaches to HOI detection have been proposed to follow this paradigm but their method for proposal generation (i.e. detecting individual objects and pairing them) still seem naive and brittle. The approach proposed in this paper fills in this gap and provides a more elegant solution (by generating pair proposals in one go) using a Transformer architecture. In this sense, the proposed approach can be viewed as a Faster R-CNN counterpart for HOI detection.
- The benchmark results are remarkable. The paper has reported significant boosts in mAP against SOTA methods on both datasets (Tab. 2 and 3). E.g., from 29.07 to 32.07 on HICO-DET (Tab. 2) and from 58.8 to 63.91 on V-COCO (Tab. 3). This is impressive given the popularity of these datasets in publications recently.
- The writing has done a good job in introducing the background and motivating the proposed approach.
- Ablation is provided for justifying various new components in the pipeline.

### Weaknesses
- Clarity in Sec. 3.4 ("dynamics re-weighting") should be improved. The technical details are difficult to follow. For example, there are two queues (L177-179) (one for object and one for action), but there is only one set of weights in Eq. (4). Also there seems to be some connections from these weights to the loss but that connection is unclear. Finally, it will be helpful to first explain at an intuitive level how these weights are used in training.
- Missing details on the formulation of various training losses (i.e. $L_b$, $L_{GIoU}$, $L_p$, $L^o_c$, and $L^a_c$) in Sec 3.5. This piece of information is important for reproducibility and thus cannot be omitted.
- Following the last point, how are the ground truths of interaction scores defined (for $L_p$ in L197)?

### Suggestions:
- [L155-157] "... to predict human bounding-box, object bounding box, and object class, ..." -> Besides these, there is also an "interactive score" in the output (See. Fig. 2 middle)? Please also add some details for that here.
- [L206] "instance classification" -> "object classification"
- [L207] "interactive probability" -> Consider using a consistent term ("interactive score" in Fig. 2 and L203) to minimize confusion. Also L197: "interactive probability loss".
- [L297] "10.08% and 9.32% point mAP gains" -> should be "relative mAP gains"?
- [Tab. 2 and 3] For the completeness of SOTA comparison, please add the benchmark results from [14] and [30] to Tab. 2 and 3.
- I am aware that CVPR'21 papers are officially published after the NeurIPS'21 deadline. But just a suggestion to add [1] below to the references in the next revision due to its high relevance to the technique herein. Also add their results to the SOTA comparison in Tab. 2 and 3.
- [Tab. 2] Please add a column for indicating whether the object detector has been fine-tuned on HICO-DET. For example, see Kim et al. ([1] below), [38], and [16].
- [Sec 4.5] Please add a pointer to Tab. 4 in the text.

[1] Bumsoo Kim, Junhyun Lee, Jaewoo Kang, Eun-Sol Kim, and Hyunwoo J. Kim. HOTR: End-to-End Human-Object Interaction Detection with Transformers. In CVPR, 2021.

### Justification of Rating
The paper has made solid technical contributions with a novel Transformer based architecture. The experimental results also show strong promise. Although there are some issues on clarity and missing information, this is still overall a solid submission.

**Time Spent Reviewing:**

7

---

> ### Author Response · Authors · 2021-08-10
> **The Response to Reviewer ez6V**
>
> We sincerely thank you for your effort and valuable comments.
>
> ---
>
> **Q1**: Clarity in Sec. 3.4 ("dynamics re-weighting") should be improved. The technical details are difficult to follow. For example, there are two queues (L177-179) (one for object and one for action), but there is only one set of weights in Eq. (4). Also, there seem to be some connections from these weights to the loss but that connection is unclear. Finally, it will be helpful to first explain at an intuitive level how these weights are used in training.
>
> **A1**: Thank you for the kind suggestion, and we will rewrite the dynamic re-weighting part in a clearer manner.
>
> In Eq. (4), we merge two formulas in one to save space, where the “(o, a)” means either o(object) or a(action), and this may cause confusion. We will separate it into two formulas. There are two sets of weights given by Eq. (4), one for the object class and one for the action class. The weights are used to the losses traditionally, i.e., re-weighting loss of each class via multiplying a coefficient calculated by Eq. (4) and then calculating the summation. The re-weighting operation is utilized on object classification and action classification separately.
>
> ---
>
> **Q2**: Missing details on the formulation of various training losses (i.e. $L_b$, $L_{GIoU}$, $L_p$, $L_{co}$, and $L_{ca}$) in Sec 3.5. This piece of information is important for reproducibility and thus cannot be omitted.
>
> **A2**: We will clarify this point in the revised version. Here, we strictly follow the losses and hyper-parameter settings of QPIC for a fair comparison.
>
> ---
>
> **Q3**: Following the last point, how are the ground truths of interaction scores defined (for $L_p$ in L197)?
>
> **A3**: The ground truths of the interaction score are converted from the HOI labels. Formally, it is a binary classification score (0 or 1) to determine whether the human-object pair is an interactive pair or not.
>
> ---
>
> **Q4**: [L155-157] "... to predict human bounding-box, object bounding box, and object class, ..." -> Besides these, there is also an "interactive score" in the output (See. Fig. 2 middle)? Please also add some details for that here.
>
> **A4**: Thank you for the kind suggestion. We will add some details about the "interactive score" in the revised version. In short, we use an interactive score head to simply determine whether the human-object pair is an interactive pair or not by a binary classification.
>
> ---
>
> **Q5**: Typos
>
> **A5**: Thanks very much for the detailed suggestions. We will further polish our paper according to these suggestions in the revised version, and we will add the suggested CVPR’21 paper into our references and comparison tables.

---

### Official Review · Reviewer_LXvb · 2021-07-15

**Rating:** 7
**Confidence:** 5

**Summary:**

This paper explored the HOI detection problem and aimed to mine the benefits of two-stage and one-stage HOI detection methods. To this end, this paper firstly gave a detailed analysis of conventional one-stage and two-stage HOI detection methods and concluded the advantages and disadvantages of the conventional methods. Then, based on their analysis, this paper proposed a Cascade Disentangling Network (CDN), which is a one-stage framework with disentangling human-object detection and interaction classification in a cascade manner. Finally, this paper conducted experiments on HICO-Det and V-COCO datasets and achieved state-of-the-art performance.

**Limitations And Societal Impact:**

yes


**Main Review:**

+ The motivation and the proposed method are clear and reasonable. With the development of one-stage and two-stage HOI detection methods, mining their advantages is the next essential step for the community.

+ This paper introduced a complete and accurate analysis about conventional HOI detection methods, and proposed a simple yet efficient solution to mine the benefits of one-stage and two-stage detector. The proposed method has achieved SOTA on two HOI detection benchmarks.

+ The results in Table 1 show that HO-PD makes a significant improvement (mAP from 15.37 to 24.05), which confirms the author's claim that one-stage methods are superior in human-object pair generation. I believe this will give some useful inspiration to the HOI field.

+ The source code is provided.

- The Dynamic Re-weighting process seems a bit complicated, hope to get a more easy-to-understand description.



**Time Spent Reviewing:**

5 hours

---

> ### Author Response · Authors · 2021-08-10
> **The Response to Reviewer LXvb**
>
> We sincerely thank you for your effort and valuable comments.
>
> ---
>
> **Q1**: The Dynamic Re-weighting process seems a bit complicated, hope to get a more easy-to-understand description.
>
> **A1**: Thanks a lot for your suggestion. We will rewrite the Dynamic Re-weighting part in a clearer manner. The general idea is to re-weight the object and action classification training since both suffer from severe long-tail class distributions. To alleviate this, we freeze parameters of a trained model with normal losses and finetune it by only training the two cascade decoders with dynamic re-weighting losses with a lower learning rate. We use two separate queues for object and action, which are used as memory banks to accumulate the number of each object class or action class in a sliding window (length: $L_Q$) during training.

---

> > ### Comment · Reviewer_LXvb · 2021-08-22
> > **My comments have been addressed.**
> >
> > After reading the authors' responses, my main concern about the unclarity of re-weighting loss is solved. Furthermore, I carefully read the other reviews and all reviewers recognize the core contribution of this paper. Therefore, I tend to keep my initial score and think that this paper can be accepted.

---

> > > ### Author Response · Authors · 2021-08-30
> > > **The Response to Reviewer LXvb**
> > >
> > > Thanks for your response and appreciation. We will update the contents of our rebuttals and provide a more straightforward description of the re-weighting module in the revised version. Thank you again for your effort and valuable suggestions.

---

### Official Review · Reviewer_5kHf · 2021-07-17

**Rating:** 6
**Confidence:** 3

**Summary:**

The paper proposes a novel HOI detection pipeline which includes a human-object pair decoder and interaction decoder once visual feature is extracted. Both the human-object pair decoder and interaction decoder are modeled by a transformer architecture composed of multiple layers of attention. It also presents some techniques for further improvement such as dynamic re-weighting and pair-wise NMS (PNMS). The proposed method presents superior detection results on common benchmarks, namely HICO-Det and V-COCO.

**Limitations And Societal Impact:**

The authors discussed the potential negative societal impact of the work. However, I didn't see the limitations being discussed.

**Main Review:**

Pros:

+ The analysis and comparison between two-stage and one-stage HOI detector is clear and reasonable. This section greatly motivates the proposed method.

+ The proposed method presents superior detection results on common benchmarks, namely HICO-Det and V-COCO. The improvement on rare classes is the most significant.

Cons:

- The title does not clearly convey the main message of the article.

-  What is "co-attention" is L148 and L166? Is it cross-attention?

- The reweighing formula is not clear. Does the superscript "(o, a)" mean either o(object) or a(action)? It will be better to focus on one at a time especially when formally presenting equation. Also, I don't understand what is "a query Q_o with length L_Q^o" and how is it integrated into the whole pipeline.

- Number of parameters and computation cost introduced by the transformer decoder?

- In Table 1, I think it is more suitable to list the results of CDN-S w/o re-weighing and PNMS for fair comparison with QPIC.

**Time Spent Reviewing:**

3 hours

---

> ### Author Response · Authors · 2021-08-10
> **The Response to Reviewer 5kHf**
>
> The Response to Reviewer 5kHf
>
> We sincerely thank you for your effort and valuable comments.
>
> ---
>
> **Q1**: The title does not clearly convey the main message of the article.
>
> **A1**: The main contribution of our work is to conduct a detailed analysis of two conventional HOI detection paradigms, i.e., two-stage and one-stage. Based on the analysis, we propose to disentangle the decoder of a one-stage HOI detector into two cascade decoders. Thus, our proposed CDN keeps the advantages of conventional one-stage methods, directly and accurately locating the interactive human-object pairs and bring the advantages of two-stage methods into our one-stage framework disentangling human-object detection and interaction classification. Therefore, the title summarizes the main message of this work, which is to mine the benefits of both two-stage and one-stage HOI detections to design a better HOI detector.
>
> ---
>
> **Q2**: What is “co-attention” in L148 and L166? Is it cross-attention?
>
> **A2**: This is indeed a cross-attention between queries and the sequenced visual features by the encoder. Here we follow the transformer-based object detector DETR, which use a multi-head co-attention mechanism.
>
> ---
>
> **Q3**: The reweighing formula is not clear. Does the superscript “(o, a)” mean either o (object) or a (action)? It will be better to focus on one at a time especially when formally presenting an equation. Also, I don’t understand what is “a query $Q_o$ with length $L_Q^o$” and how is it integrated into the whole pipeline.
>
> **A3**: In Eq.(4), the “(o, a)” means either o(object) or a(action). We merge two formulas in one to save space since the only difference is this symbol. We will separate it into two formulas in the revised version.
>
> We apply two similar queues, one for the object (o) and another for action (a). The length of the queue is $L_Q^o$ and $L_Q^a$, respectively. Therefore, the queues are used as a memory bank to accumulate the number of each object class or action class in a sliding window (length: $L_Q$) during training.
>
> ---
>
> **Q4**: Number of parameters and computation cost introduced by the transformer decoder?
>
> **A4**: The number of parameters and flops of CNS-S is 39.534 M and 24.02 G for the input size of HxW=512x512. They are the same as QPIC since they both have a six-layers decoder (for CDN-S, it has two 3-layer cascaded decoders).
> For CDN-B, the number of parameters is 48.567 M. The additional 6 more layers of the cascaded decoder introduce 9.033 M parameters and 0.463G flops for the input size of HxW=512x512. Thus the extra computation cost is limited.
>
> ---
>
> **Q5**: In Table 1, I think it is more suitable to list the results of CDN-S w/o re-weighting and PNMS for a fair comparison with QPIC.
>
> **A5**: Thanks for the suggestion. We will add CDN-S w/o re-weighting and PNMS in Table 1. In this case, the mAPs are 30.96 for “Full”, 27.02 for “Rare”, and 32.14 for “Non-Rare”, which still outperform QPIC (29.07 for “Full”, 21.85 for “Rare” and 31.23 for “Non-Rare”) by a large margin.

---

### Decision · Program_Chairs · 2021-09-27

**Decision:**

Accept (Poster)

**Comment:**

This paper presents work on human-object interaction (HOI) detection.  The main contributions include an analysis of two-stage versus one-stage HOI detection methods and a multi-layer transformer architecture that achieves state of the art results.  The initial reviews pointed to concerns over clarity of some terminology and overall originality and significance of the contributions.  However, the reviewers unanimously recommended acceptance of the paper after the discussion period, based on the good empirical results and sensible architecture.